# SpeechEE: A Novel Benchmark for Speech Event Extraction

Bin Wang
Harbin Institute of Technology
(Shenzhen)
Shenzhen, China
23s051047@stu.hit.edu.cn

Meishan Zhang
Harbin Institute of Technology
(Shenzhen)
Shenzhen, China
zhangmeishan@hit.edu.cn

Hao Fei*
National University of Singapore
Singapore, Singapore
haofei37@nus.edu.sg

Yu Zhao
Tianjin University
Tianjin, China
zhaoyucs@tju.edu.cn

Bobo Li
Wuhan University
Wuhan, China
boboli@whu.edu.cn

Shengqiong Wu
National University of Singapore
Singapore, Singapore
swu@u.nus.edu

Wei Ji
National University of Singapore
Singapore, Singapore
jiwei@nus.edu.sg

Min Zhang
Harbin Institute of Technology
(Shenzhen)
Shenzhen, China
zhangmin2021@hit.edu.cn

## Abstract

Event extraction (EE) is a critical direction in the field of information extraction, laying an important foundation for the construction of structured knowledge bases. EE from text has received ample research and attention for years, yet there can be numerous real-world applications that require direct information acquisition from speech signals, online meeting minutes, interview summaries, press releases, etc. While EE from speech has remained under-explored, this paper fills the gap by pioneering a **SpeechEE**, defined as detecting the event predicates and arguments from a given audio speech. To benchmark the SpeechEE task, we first construct a large-scale high-quality dataset. Based on textual EE datasets under the sentence, document, and dialogue scenarios, we convert texts into speeches through both manual real-person narration and automatic synthesis, empowering the data with diverse scenarios, languages, domains, ambiences, and speaker styles. Further, to effectively address the key challenges in the task, we tailor an E2E SpeechEE system based on the encoder-decoder architecture, where a novel Shrinking Unit module and a retrieval-aided decoding mechanism are devised. Extensive experimental results on all SpeechEE subsets demonstrate the efficacy of the proposed model, offering a strong baseline for the task. At last, being the first work on this topic, we shed light on key directions for future research. Our codes and the benchmark datasets are open at https://SpeechEE.github.io/.

## CCS Concepts

• **Information systems** → **Multimedia information systems**.

---

*Hao Fei is the corresponding author.

---

## Keywords

Information Extraction, Event Extraction, Speech Modeling, Spoken Language Understanding

**ACM Reference Format:**
Bin Wang, Meishan Zhang, Hao Fei, Yu Zhao, Bobo Li, Shengqiong Wu, Wei Ji, and Min Zhang. 2024. SpeechEE: A Novel Benchmark for Speech Event Extraction. In *Proceedings of the 32nd ACM International Conference on Multimedia (MM '24), October 28-November 1, 2024, Melbourne, VIC, Australia.* ACM, New York, NY, USA, 10 pages. https://doi.org/10.1145/3664647.3680669

## 1 Introduction

Event extraction [5, 40] is a critical task within the information extraction community [13, 29], aimed at automatically identifying structured information from various data sources. It seeks to delineate the semantic structure encapsulating the essence of '*who or what does what to whom, when, where, and why*' [8]. Initially, research in EE predominantly focuses on textual data [60]; and over time, it became evident that events could be conveyed through a myriad of modalities and information sources. Subsequently, the scope of EE has broadened to include more diverse modalities and information sources, leading to significant strides in extracting events from images [32] and even videos [4]. Despite these advances, extracting events from speech or audio signals remains a largely under-explored topic. We argue that EE in speech also holds immense research significance and practical value, given its applicability in a variety of real-life scenarios, including meetings, lectures, interviews, and news reports, especially in scenarios where transcription is not available.

In response to this gap, in this paper, we introduce a novel task: Speech Event Extraction (namely, **SpeechEE**). SpeechEE is designed to process audio inputs and output structured event records, identifying event triggers, categorizing event types, recognizing arguments and classifying their roles. To benchmark this task, we develop a large-scale and high-quality comprehensive dataset. On the one hand, based on 8 common textual EE datasets under the *sentence*, *document* and *dialogue* upon ACE EE annotation format [8],

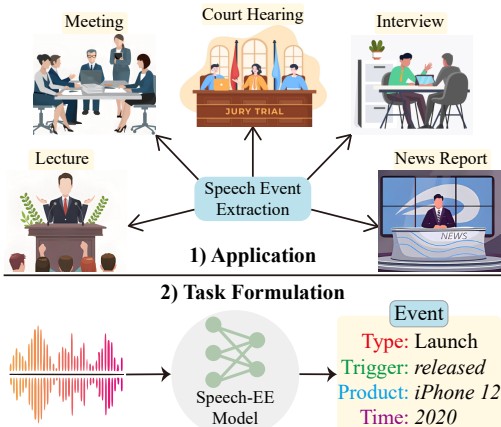

**Figure 1: An illustration of the speech event extraction's broad applications and task formulation.**

we meticulously convert texts into diverse and authentic speeches through manual real-person narration, where we simulate environments with both quiet and noisy backgrounds. To further enlarge the data quantity, we automatically synthesize the SpeechEE data via state-of-the-art (SoTA) text-to-speech (TTS) systems, progressively extending the amount while preserving all the characteristics. Strict human cross-inspection is conducted to ensure the high quality of the whole speech data. The final SpeechEE dataset comprises 8 subsets and above 260 hours of audio, featuring diverse 1) **scenarios** (sentences, documents, dialogues), 2) **languages** (English and Chinese), 3) **domains** (news, cybersecurity, movies, etc.), 4) **ambiences** (noisy and quiet) and 5) **speaker styles**. Our experimental analyses reveal that the SpeechEE dataset poses greater challenges compared to traditional textual EE, underscoring the complexity and uniqueness of speech as a medium for EE.

Modeling SpeechEE presents indispensable challenges. The most straightforward approach is first converting speech to text using Automated Speech Recognition (ASR) tools [2], followed by applying existing textual EE techniques [3]. However, this pipeline approach suffers from significant error propagation issues. More crucially, it fails to address several key bottlenecks inherent to SpeechEE. **First**, speech inherently flows without clear word boundaries, presenting a challenge in accurately identifying the precise audio segments associated with event triggers and arguments. **Second**, in real scenarios, speech might encompass background noise, hindering the effective extraction of event-relevant semantic features (e.g., triggers and arguments) directly from the characteristics of the speech itself. **Third**, the length of audio signals can be significantly greater than their text counterparts, often by orders of magnitude, adding complexity to the modeling of speech-to-event extraction. **Finally**, the presence of homophones and near-homophones in speech can lead to inaccuracies in entity recognition. For instance, words like 'peace' vs. 'piece' or 'male' vs. 'mail' can lead to inaccuracies, particularly with uncommon nouns, such as rare names or locations. Capturing these nuances accurately poses a significant challenge for models.

To effectively tackle these challenges, we propose a novel E2E model for SpeechEE task, which has been shown in Fig. 2. First, our model leverages an encoder-decoder generative framework

[37] to directly produce target event schemes from speech, thereby avoiding the need for piecemeal speech segmentation. Then, we employ contrastive learning [25] at the encoder stage for event representation learning, enhancing the model's ability to discern and disentangle event semantics from speech features. Further, a Shrinking Unit module is designed to alleviate the disparities in modal length between speech and text through projection and downsampling techniques. Finally, our model incorporates a retrieval-aided decoder that leverages an external Entity Dictionary, enabling flexible decision-making during decoding to generate new tokens or retrieve entities directly from the dictionary. Our experiments on the SpeechEE dataset demonstrate that our model outperforms pipeline baseline systems consistently. Further analysis confirms the substantial contributions of all the proposed components in effectively modeling speech for EE.

In summary, our contributions are threefold:
• We for the first time pioneer a novel task, SpeechEE, for extracting events from speech, along with the first large-scale high-quality dataset encompassing multiple scenarios, domains, languages, ambiences, and styles from both synthesis and manual crafting.
• We propose an E2E SpeechEE model that addresses key challenges in SpeechEE and offers a strong baseline performance on the benchmark data.
• We outline potential future directions to further advance the field of SpeechEE, setting the stage for ongoing research and development in this promising area.

## 2 Related Works

EE is one of the kernel subtopics within the field of information extraction [1, 5, 17, 22, 34, 41], and has been the subject of extensive research for many years. The majority of EE research has focused on textual EE [3, 37] due to the abundance of text information available online. As one of the most popular tasks in natural language processing, EE has evolved over decades and garnered significant research attention. A variety of EE methods have been proposed, such as classification approaches [5], sequence labeling techniques [61], question-answering formats [28], and generation methods [20], etc. Given its critical applications across numerous scenarios, EE has been integrated into various modalities [4, 32], such as image and video EE [53, 63]. On the other hand, while research on Named Entity Recognition (NER) [18] and Relation Extraction (RE) [43, 44, 58] under text and speech have emerged within the IE community, research specifically addressing EE under speech remains conspicuously absent. Thus, this work endeavors to fill this gap by establishing a comprehensive SpeechEE benchmark dataset. We develop a large-scale dataset through both manual and automated methods, broadly encompassing multiple scenarios, languages, domains, ambiences, and speaker styles.

This work also relates to the speech modeling research [6, 7, 49]. Audio signals represent a significant modality in the world, especially within human society where speech is a primary mode of communication, which is one of the key research tracks within the multimodal learning communities [10, 14, 27, 30, 54], e.g., image modeling [23, 24, 31, 57], video modeling [11, 12, 39, 66]. Consequently, the speech community has focused on various tasks, directions, and research scenarios, including ASR [2], TTS [46], and Spoken Language Understanding (SLU) [47], among others. To enable

machines to comprehend the semantic information in speech, particularly to extract linguistic information such as event structures, accurate understanding of speech is required. The conventional approach [58] involves first using ASR technology to transcribe speech into text, followed by the application of established NLP techniques for textual analysis. However, this pipeline approach inevitably introduces distortions in information extraction from the given speech due to potential errors in ASR, thereby incorporating noise [38]. In this paper, we construct an E2E SpeechEE model that addresses a series of unique challenges associated with SpeechEE.

## 3 Task Definition of SpeechEE

We mainly follow the ACE scheme [8] for the event definition. We formulate the SpeechEE task as: given a speech audio consisting of a sequence of acoustic frames $S = (f_1, f_2, \cdots, f_U)$, the pre-defined event type set $E$ and argument role set $R$, we aim to extract all possible structured event records consisting of four parts: 1) event type $\varepsilon \in E$, 2) event trigger, 3) event argument role $r \in R$ and 4) event argument.

## 4 Benchmark Data Construction

### 4.1 Construction Approach

**Data Source.** We consider constructing our SpeechEE data based on existing textual EE benchmark datasets, since textual EE datasets are well-defined and well-established, with readily available and accurate event annotations. Specifically, we utilize datasets from three scenarios: 1) sentence-level data, including ACE05-EN$^+$ [35], ACE05-ZH [52], PHEE [48], CASIE [45], and GENIA [26]; 2) document-level data, RAMS [9] and WikiEvents [33]; and 3) dialogue-level data, Duconv subset in CSRL [59]. Each dataset strictly follows the ACE EE schema and is annotated with corresponding EE records.

**Manual Speech Narration.** Based on the above textual data, we then obtain the corresponding speech signals through manual reading. Note that each dataset also maintains its original train/dev/test split, which we do not alter. For each dataset's respective languages, we employ 10 native speakers of different genders and ages to ensure speech diversification in terms of tone and timbre. Each speaker is instructed to read the original text data in both quiet and noisy background settings, such as in a cafeteria, meeting room, street, and classroom, to cover as many real-life scenarios as possible. To ensure the quality of the acquired speech, we conduct manual cross-inspection. Specifically, 2 individuals simultaneously listen to the same speech recording and score it from 1-10 (low to high quality) based on the audio's accuracy in reflecting the original text. Finally, we calculate the Cohesion Kappa score across the 2 auditors, retaining only instances where the score exceeded **0.85**, thereby ensuring high consistency in data quality.

**Automatic Synthesis.** Due to the huge workload and cost of manual speech recording, as well as the fact that some scenarios has hard real environment reproduction, we can record only a portion of speech for each original text dataset. To substantially expand our SpeechEE dataset, we now consider using automatic synthesis to continue building speech. Note that we only enrich the train set of SpeechEE data. Our basic idea is using TTS tools to convert text to speech automatically, during which we strictly control the

**Table 1: Key characteristics of our SpeechEE dataset.**

| Scenario | Source | Language | Domain | Tone | Event-Type | Arg.-Role |
|---|---|---|---|---|---|---|
| Sentence | ACE05-EN$^+$ | English | News | 10 | 33 | 22 |
| | ACE05-ZH | Chinese | News | 6 | 33 | 27 |
| | PHEE | English | Medical | 10 | 2 | 16 |
| | GENIA | English | Biology | 10 | 5 | - |
| | CASIE | English | Cyber | 10 | 5 | 26 |
| Document | RAMS | English | News | 7 | 139 | 65 |
| | WikiEvents | English | General | 7 | 50 | 59 |
| Dialogue | Duconv | Chinese | Movies | 6 | 1 | 8 |

**Table 2: Statistics of the SpeechEE dataset. In the brackets are the splits of train/develop/test sets.**

| Scenario | Source | Human | Synthesis |
|---|---|---|---|
| Sentence | ACE05-EN$^+$ [35] | 1,000 (800/100/100) | 12,867 |
| | ACE05-ZH [52] | 1,000 (800/100/100) | 6,311 |
| | PHEE [48] | 500 (400/50/50) | 2,898 |
| | GENIA [26] | 1,000 (800/100/100) | 15,023 |
| | CASIE [45] | 500 (400/50/50) | 3,751 |
| | Total | 4,000 | 40,850 |
| Document | RAMS [9] | 1,000 (800/100/100) | 7,329 |
| | WikiEvents [33] | 100 (80/10/10) | 206 |
| | Total | 1,100 | 7,535 |
| Dialogue | Duconv [59] | 140 (100/20/20) | 1,200 |

synthesized voice's timbre and ambient sounds. We mainly use two high-performance open-source TTS tools: Bark[1] and Edge-TTS[2]. Notably, we perform denoising of the original text before auto-recording to filter out instances that are inexpressible in speech or irrelevant to the task, thus ensuring the quality of the resulting speech. To ensure the quality of the synthesized data, we consider different evaluation methods from those used for manual construction. Specifically, we evaluate from both objective and subjective perspectives: the former assesses the text accuracy of synthesis speech, and the latter evaluates such as the naturalness, timbre, and accuracy of ambient sounds of the speech. Objectively, we use an ASR model to evaluate the word error rate of the synthesis speech. Subjectively, we employ two native speakers to rate the speech on a 10-scale, retaining only instances where the score exceeds **0.8**.

### 4.2 Data Insights and Characteristics

The total duration of the final SpeechEE data is above 260 hours. The data characteristics are clearly illustrated in Table 1. And in Table 2 we display the detailed statistics. Following we summarize the highlights of SpeechEE dataset.

● **Multiple Scenarios:** SpeechEE covers three major common scenarios of existing EE: sentence, document and dialogue.

● **Diverse Domains:** SpeechEE involves diverse domains, such as news, biomedicine, cybersecurity, movie fashions, etc.

● **Multilinguity:** Datasets in SpeechEE cover two languages, English in 6 subsets and Chinese in 2 subsets.

[1]https://github.com/suno-ai/bark
[2]https://github.com/rany2/edge-tts

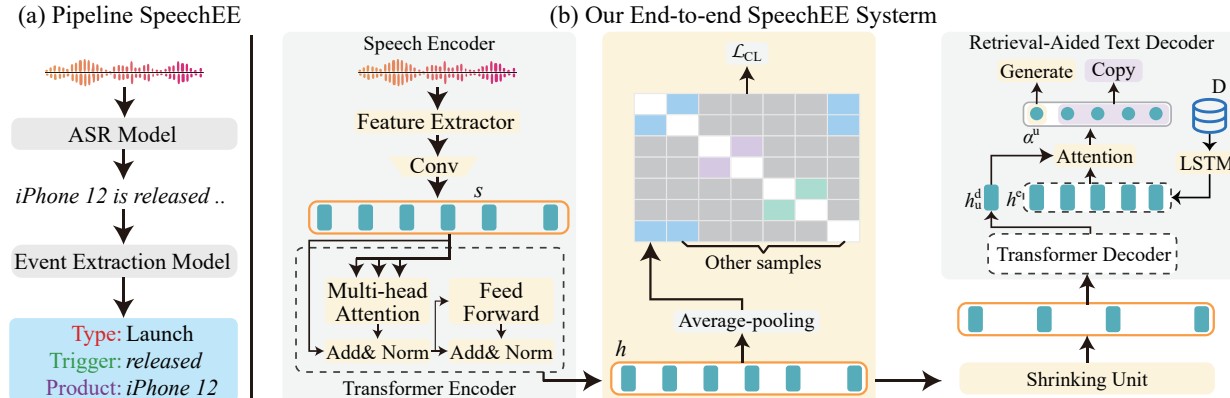

**Figure 2: The architecture of the pipeline and E2E SpeechEE model.**

• **Various Ambiences:** Speeches either have quiet backgrounds or noisy backgrounds. The noisy background setting covers many scenarios that simulate realistic environments of the task.

• **Rich Styles:** SpeechEE features a diverse range of human voice styles, e.g., male, female, and child voices, and also different speaker tones and timbres.

• **Large Scale and High Quality:** We create a large-scale dataset through both manual annotation and automatic synthesis, comprising over 260 hours of audio. The quality has been rigorously controlled through strict cross-validation.

## 5  SpeechEE Method

Now we introduce two methods to address SpeechEE, including pipeline SpeechEE and E2E SpeechEE. The pipeline SpeechEE is a two-step method that firstly uses ASR system to obtain the transcripts of input speech and then uses textual EE model to extract event records from transcripts. We then propose an E2E SpeechEE model to extract the event records from speech in one shot. Two SpeechEE architectures are overviewed in Fig. 2.

### 5.1  Pipeline Model

The straightforward way for SpeechEE is to divide it into two subtasks, ASR and textual EE, and cascade strong-performing off-the-shelf models as a two-step pipeline, shown in Fig. 2(a). Here we provide a feasible implementation. We first employ the Whisper [42] model as the ASR module to convert speech signals to the corresponding transcripts. Compared with other ASR tools (Wav2Vec 2.0 [2] and HuBERT [21]), the Whisper model is trained on a considerably labeled audio corpus, and thus can directly learn the mapping from speech to text, with more superior speech recognition performance than other ASR models. Additionally, the whisper model incorporates data from multiple languages and domains, which matches the multilingual and diverse domain characteristics of the SpeechEE dataset we construct. For TextEE module, we adopt Text2Event [37] which is a generative-based E2E EE method. Text2Event features a sequence-to-structure paradigm, which can directly perform textual EE based on the whisper model outputs. This also helps tackle the lack of fine-grained annotations about the boundary of event trigger and argument mention in SpeechEE.

### 5.2  End-to-End Model

As mentioned previously, the pipeline SpeechEE method faces significant error propagation issues. To address this, we propose an end-to-end (E2E) approach. As illustrated in Fig. 2(b), the overall framework has an encoder-decoder structure, which mainly consists of a speech encoder, a Shrinking Unit module, and a retrieval-aided text decoder.

**Speech Encoder.** We take the speech encoder as in the Whisper model, which is built based on an acoustic feature extractor and a normal transformer encoder. The input speech $S$ is firstly processed by an acoustic feature extractor to get an 80-channel log-magnitude Mel spectrogram clip sequence. Then a small stem consisting of two convolution layers with a filter width of 3 and the GELU activation function [19] is applied to transfer the clip sequence to the inputs of transformer $s$. Afterward, the transformer encoder encodes the spectrogram representation into hidden states $\boldsymbol{h} = (\boldsymbol{h}_1, \boldsymbol{h}_2, \cdots, \boldsymbol{h}_T)$, where $T$ is the sequence length of hidden states $\boldsymbol{h}$. We use the last encoder layer's hidden vectors as speech representations.

$$\boldsymbol{s} = \text{Conv}(\text{FeatureExtractor}(S)), \qquad (1)$$

$$\boldsymbol{h} = \text{TransformerEncoder}(\boldsymbol{s}). \qquad (2)$$

The speech representations can adequately capture the acoustic feature due to the ASR pre-training of Whisper. However, these speech representations are not capable of modeling the event-related features for the SpeechEE task. Thus, we design a contrastive learning strategy to enhance speech representation. To better learn the event-related semantics, we choose the positive and negative samples based on the event type. For the same event type, the encoded speech representations will be pulled together while representations for different event types should be pushed away. For a batch of $N$ samples, the contrastive learning loss $\mathcal{L}_{\text{CL}}$ is defined as follows:

$$\mathcal{L}_{\text{CL}} = -\sum_{i=1}^{N} \log \frac{\exp\left(\boldsymbol{x}^i \cdot \boldsymbol{x}^+ / \tau\right)}{\sum_{\boldsymbol{x} \in K} \exp\left(\boldsymbol{x}^i \cdot \boldsymbol{x} / \tau\right)}, \qquad (3)$$

where $K$ is a set composed of all samples in the batch, $\boldsymbol{x}^+$ denotes the positive sample, and $\tau$ is the temperature hyper-parameter. The $\boldsymbol{x}^i$ is the average pooling results for the i-th sample's speech representation $\boldsymbol{h}$.

**Shrinking Unit.** Speech sequences are usually much longer than corresponding text sequences, and there exists even more redundant information for the EE task. The discrepancy in sequence length of different modalities adds great difficulty to the modeling of SpeechEE task. To combat this, we here propose a novel Shrinking Unit module, which is added between the speech encoder and text decoder in our E2E architecture, to mitigate the difference in the sequence length by projection and downsampling. Technically, $n$ one-dimensional convolutional layers downsample the encoder output $h$ using a stride of $m$, which can shorten the sequence by a factor of $m^n$.

$$h_s = \text{ShrinkingUnit}(h) . \tag{4}$$

**Retrieval-Aided Text Decoder.** After the hidden state vectors are filtered by the Shrinking Unit, a pre-trained text decoder predicts the output event structure token-by-token with the sequential input tokens' hidden vectors. At the step $k$ of generation, the text decoder predicts the $k$-th token $y_k$ and decoder state $h_k^d$ as follows:

$$h_k^d, y_k = \text{Decoder}(h, h_1^d, \cdots, h_{k-1}^d, y_{k-1}) . \tag{5}$$

Yet due to the phenomenon of homophones and near-sound words in speech, it is hard for the decoder to precisely extract some entity mentions, especially for rarely-seen words during the training. Inspired by the Contextual ASR [65], we propose to incorporate a retrieval mechanism and leverage an external Entity Dictionary to flexibly decide whether to generate a new token or retrieve an existing entity from the Entity Dictionary. The retrieval mechanism can help to constrain the difficult entities with a closed set and avoid generating incorrect ones with flawed homophones and near-sound words.

Technically, the Entity Dictionary is constructed with entities that appear only once in the training set, covering the rarely-seen words that are hard to recognize. We assume that we have no other prior knowledge about the test data. Therefore, the train set from the same origin can be used as the external knowledge properly. We denote the Entity Dictionary as $D = \{e_0, e_1, e_2, \cdots, e_l\}$, where $e_0$ is added to note the no-entity option. The Entity Dictionary is firstly encoded by an LSTM module to get the last state as the fixed dimensional entity representation $h^e = \{h_0^e, h_1^e, \cdots, h_l^e\}$. Then the attention score for entity $e_j$ is computed where query denotes the last layer of decoder state $h^d$ and the key denotes the entity representation $h^e$.

$$\alpha_j^u = \frac{(W_q h_u^d) \cdot (W_k h_j^e)}{\sqrt{d_{att}}} , \tag{6}$$

where $d_{att}$ denotes the dimension of $h_j^e$, $u$ denotes the decoding step, $W_q$ and $W_k$ are two learned linear transformation parameters of query and key respectively. After softmax, we obtain the retrieved probability $p_j^u$ of the entity $e_j$. It is used to flexibly decide to retrieve which existing entity in the dictionary or generate the output by decoder. Then we compute the loss by using the golden entity label and the retrieved probability.

$$\mathcal{L}_{ED} = -\sum_u \log p_g^u , \tag{7}$$

where $g$ denotes the golden entity in time step $u$. The final loss is composed of $\mathcal{L}_{ED}$ and the contrastive loss $\mathcal{L}_{CL}$ in Equation 3.

**Table 3: Overall results on sentence-level datasets.**

| | TI | TC | AI | AC | *Avg* |
|---|---|---|---|---|---|
| ● *ACE05-EN+* | | | | | |
| Pipeline (Bart) | 60.8 | 57.0 | 33.3 | 20.2 | 42.8 |
| E2E (Bart) | 63.5 | 59.3 | 35.5 | 23.0 | 45.3 +2.5 |
| Pipeline (T5) | 61.2 | 57.1 | 33.1 | 20.4 | 43.0 |
| E2E (T5) | **65.0** | **61.1** | **35.3** | **23.2** | **46.2** +3.2 |
| ● *ACE05-ZH* | | | | | |
| Pipeline (mBart) | 42.5 | 29.7 | 18.7 | 15.8 | 26.7 |
| E2E (mBart) | 43.3 | 33.6 | 19.9 | 16.7 | 28.4 +1.7 |
| Pipeline (mT5) | 43.9 | 30.8 | 17.6 | 14.7 | 26.8 |
| E2E (mT5) | **44.2** | **34.5** | **20.1** | **17.3** | **29.0** +2.2 |
| ● *PHEE* | | | | | |
| Pipeline (Bart) | 50.1 | 49.1 | 25.9 | 23.8 | 37.2 |
| E2E (Bart) | **53.1** | **50.5** | 29.2 | 27.1 | 40.0 +2.8 |
| Pipeline (T5) | 49.4 | 47.0 | 27.9 | 25.0 | 37.3 |
| E2E (T5) | 52.6 | 49.7 | **32.2** | **29.9** | **41.1** +3.8 |
| ● *GENIA* | | | | | |
| Pipeline (Bart) | 23.5 | 20.9 | - | - | 22.2 |
| E2E (Bart) | 27.1 | 24.3 | - | - | 25.7 +3.5 |
| Pipeline (T5) | 21.1 | 18.3 | - | - | 19.7 |
| E2E (T5) | **28.1** | **25.3** | - | - | **26.7** +7.0 |
| ● *CASIE* | | | | | |
| Pipeline (Bart) | 55.2 | 54.5 | 36.6 | 32.9 | 44.8 |
| E2E (Bart) | 56.2 | 55.3 | **38.6** | **35.1** | **46.3** +1.5 |
| Pipeline (T5) | 53.2 | 52.5 | 36.0 | 31.9 | 43.4 |
| E2E (T5) | **56.5** | **56.0** | 37.9 | 34.0 | 46.1 +2.7 |

## 6 Experiments

### 6.1 Settings

We carry out experiments on our SpeechEE datasets. For the pipeline baseline, we use whisper-large-v2 as the ASR module and choose Text2Event with two different language models, i.e., Bart-large and T5-large as the TextEE module. For our E2E method, for a fair comparison, we also adopt the encoder of whisper-large-v2 as the speech encoder and use the decoder of pre-trained language models (Bart-large and T5-large) as our text decoder. In particular, we change Bart-large and T5-large to mBart-large-50 and mT5-large for Chinese subsets, ACE05-ZH and Duconv.

For efficient training, we freeze the acoustic feature extraction module of whisper and train the self-attention encoder, Shrinking Unit module, cross-attention between encoder and decoder, and Entity Dictionary attention. We train all SpeechEE models and optimize the models using AdamW [36]. We conduct all the experiments on NVIDIA A100 80GB. All of our models are evaluated on the best-performing checkpoint on the validation set. After the model inference, we need to parse the generated linear output of the structured tree to obtain the final tuples of structured event records. For evaluation, we first normalize the event records by converting them to lowercase format to avoid innocuous errors. Then we follow the same F1 metrics of four event elements as in the previous study [35, 37, 51], including Trigger Identification (**TI**), Trigger Classification (**TC**), Argument Identification (**AI**) and Argument Identification (**AC**).

### 6.2 Main Results of SpeechEE

We present the main comparisons on different datasets under three scenarios, in Table 3, 4 and 5, respectively. We note that these are testing results, where the models are trained on the mixture of

**Table 4: Overall results on document-level datasets.**

|  | TI | TC | AI | AC | *Avg* |
|---|---|---|---|---|---|
| • *RAMS* | | | | | |
| Pipeline (Bart) | 76.2 | 32.6 | 18.8 | 17.3 | 36.2 |
| E2E (Bart) | **77.4** | 36.7 | 20.9 | 19.3 | 38.6 +2.4 |
| Pipeline (T5) | 76.5 | 33.7 | 20.0 | 18.2 | 37.1 |
| E2E (T5) | 76.8 | **37.2** | **21.8** | **20.6** | **39.1** +2.0 |
| • *WikiEvents* | | | | | |
| Pipeline (Bart) | 32.1 | 29.0 | 14.0 | 10.8 | 21.5 |
| E2E (Bart) | 35.3 | 33.6 | 17.4 | 14.2 | 25.1 +3.6 |
| Pipeline (T5) | 32.3 | 28.8 | 12.8 | 9.9 | 21.0 |
| E2E (T5) | **35.8** | **34.0** | **18.0** | **16.0** | **26.0** +5.0 |

**Table 5: Results on dialogue-level dataset. Duconv dataset has only one event type, thus no TC evaluation.**

|  | TI | TC | AI | AC | *Avg* |
|---|---|---|---|---|---|
| • *Duconv* | | | | | |
| Pipeline (mBart) | 42.3 | - | 22.4 | 20.2 | 28.3 |
| E2E (mBart) | 45.4 | - | 23.8 | 20.7 | 30.0 +1.7 |
| Pipeline (mT5) | 43.1 | - | 22.2 | 19.8 | 28.4 |
| E2E (mT5) | **45.9** | - | **24.0** | **21.1** | **30.3** +1.9 |

the human-reading and synthesis training sets. According to the results, three key general observations can be found. First, we see that our E2E SpeechEE method consistently achieves overall better results than the pipeline baseline among all different-level datasets. Besides, there is an effect of different pre-trained language models on the performance of SpeechEE, where T5 has stronger effects than Bart in most cases. Lastly, for the four elements of EE, the recognition and classification of argument tend to be significantly lower than those of trigger, demonstrating the greater challenges faced by the Event Argument Extraction (EAE) task. Following we summarize the detailed scenario-specific observations.

**Sentence-level Performance.** On ACE2005 dataset, both the pipeline and E2E methods exhibit a significant performance disparity between the Chinese and English datasets. This is possibly due to that, 1) ASR model indeed performs better on English data than other languages; or 2) the performance of the language model differs in language as well.

**Document-level Performance.** From the Table 4, it can be seen that the RAMS dataset receives better results on the TI task, and the performance of the TC task decreases significantly, which is mainly because RAMS includes many more event types (139 types), which leads to poor performance on the TC task. The WikiEvents data contains longer documents (speech over 5 minutes) and more events&arguments with more complicated event schema, compared to RAMS dataset, where there thus are greater challenges of EE.

**Dialogue-level Performance.** The general pattern in Table 5 is mostly coincident with above results. Overall, the performance on document and dialogue levels of SpeechEE is comparatively lower, leaving more room for improvement.

## 6.3 Model Ablation Study

Here we provide ablation results on our advanced E2E system, to ground the exact contribution of each component and design within, i.e., Contrastive Learning (CL), Shrinking Unit (SU), and

**Table 6: Ablation results (F1) on PHEE dataset. SU: Shrinking Unit, CL: Contrastive Learning, ED: Entity Dictionary.**

|  | TI | TC | AI | AC | *Avg* |
|---|---|---|---|---|---|
| E2E (T5) | 52.6 | 49.7 | 32.2 | 29.9 | 41.1 |
| w/o SU | 52.1 | 49.3 | 31.5 | 29.0 | 40.5 -0.6 |
| w/o CL | 51.0 | 48.4 | 31.0 | 28.3 | 39.7 -1.4 |
| w/o ED | 51.7 | 48.7 | 29.8 | 27.6 | 39.5 -1.6 |

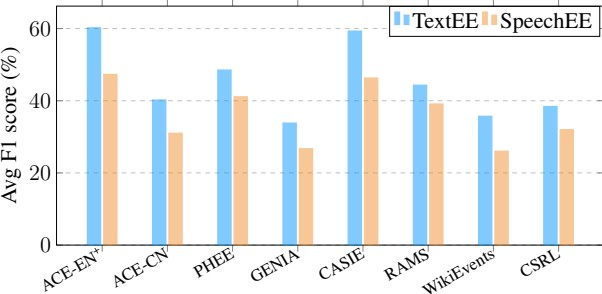

**Figure 3: Comparisons between TextEE and SpeechEE.**

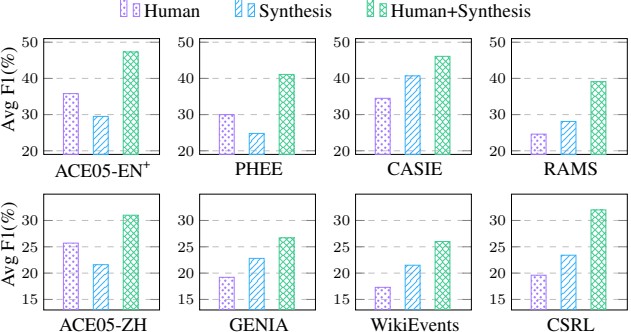

**Figure 4: Analysis of the effect of synthesis dataset.**

Entity Dictionary (ED). We representatively select the sentence-level PHEE dataset with T5-large backbone. As shown in Table 6, we see that the performance of all four event elements drops persistently when any of the three modules is removed, which shows each of them is indispensable. Specifically, the CL module gains better performance in the event detection task. This indicates that the representation learning on speech and event features is of the most importance. In contrast, the ED module plays a more role in the EAE task.

## 6.4 Analysis and Discussion

In this part, we delve deeper into our data and model, aiming to provide a more thorough understanding of them.

**Q1: Is SpeechEE Really More Challenging Than TextEE?** As a primary question, we aim to determine whether SpeechEE is meaningful and whether it presents greater challenges for the Event Extraction (EE) task compared to traditional textual EE. To this end, we compared the performance differences between SpeechEE and the corresponding TextEE. Using the same language model, T5-large, and adopting the identical generation-based E2E TextEE method across all eight datasets, the results are shown in Fig. 3. With a similar model (except for the encoder for the input signal),

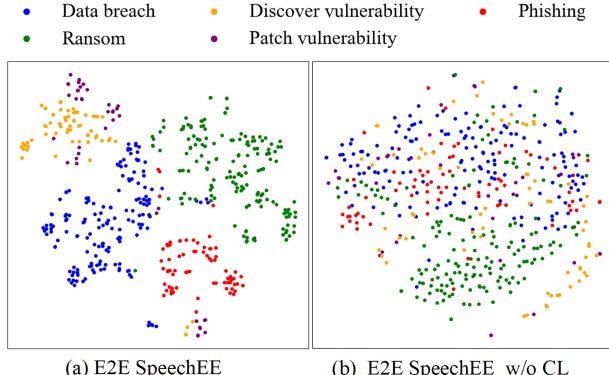

(a) E2E SpeechEE    (b) E2E SpeechEE w/o CL

Figure 5: T-SNE visualization about the effect of CL.

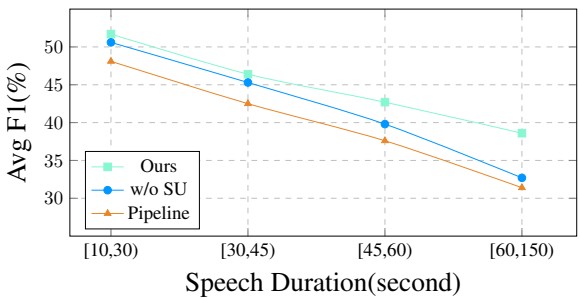

Figure 6: Impact of different speech durations on SpeechEE performance.

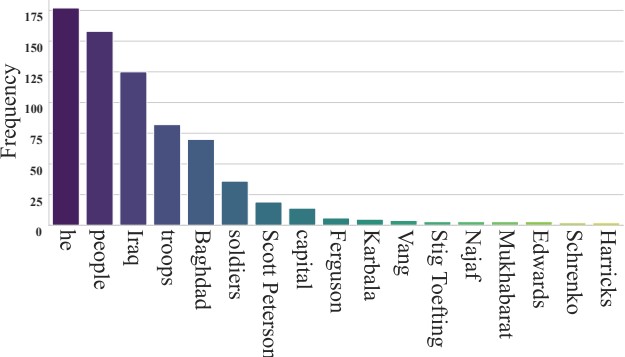

Figure 7: Frequent entities in ACE05-EN⁺ dataset.

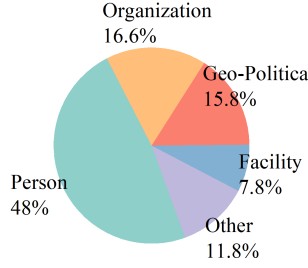

Figure 8: The component of Entity Dictionary.

the performance of SpeechEE is generally lower than that of TextEE. This highlights the real challenges faced by the SpeechEE task.

**Q2: Is It Necessary to Develop Synthesis Data?** Next, we explore whether maintaining a synthesis dataset to enlarge the training corpus is meaningful to SpeechEE. Using the E2E model, we carry out experiments on all SpeechEE subsets under 3 varied training data: 1) only real data, 2) only synthesis data, and 3) mixture of two data. The overall EE results (average F1) are shown in Fig. 4. As seen, when comparing the performance using real data and synthesis data, the former setting performs largely better than the latter, such as on ACE05-EN⁺, ACE05-ZH, PHEE, and CASIE datasets under sentence-level scenario. While for the rest datasets, the synthesis data yields better results. This is because the corresponding human-reading data are much smaller than synthesis data, where the former might not provide rich enough features for learning the pattern. Unsurprisingly, the system trained merely on human-reading data has a large decrease compared to the mixed training dataset. This directly indicates that the synthesis data is effective in relieving the data scarcity issue for SpeechEE, even though the quality of the synthesis data might be inferior to the real speech.

**Q3: How Does the Contrastive Learning Contribute?** The above ablation study has demonstrated the efficacy of the Contrastive Learning mechanism in our system, in boosting the speech and event representations. To reveal how it exactly improves the performance, here we present the visualization of the embeddings. Technically, we randomly select 500 samples from the CASIE dataset

labeled with 5 event types, and then obtain the speech encoder output embedding of them, and finally visualize the representations via t-SNE algorithm [50]. From Fig. 5, it is obvious that the samples of different event categories after Contrastive Learning have clearer boundaries than those without the mechanism, indicating that Contrastive Learning helps to learn event features from speech. In addition, we observe that among the 5 event types in the CASIE dataset, Patch-vulnerability and Discover-vulnerability show relatively poorer performance, which may be caused by the high semantic similarity between these two event types.

**Q4: How Does the Shrinking Unit Module Address Lengthy Audio Signals?** Speech signals are often much longer than text (especially in the form of long documents), which undoubtedly increases the modeling complexity of SpeechEE. We now evaluate our model's performance across different speech lengths. We consider the RAMS document data, where we grouped speech into four-length segments to observe the model's results. As illustrated in Fig. 6, as the length of the speech sequence increases, there is a significant decrease in the performance of all three systems. However, our E2E model, equipped with the full Shrinking Unit (SU) mechanism, effectively counters this trend, demonstrating its effectiveness. In contrast, our model without the SU experiences the most severe performance drop when the speech length exceeds 60 seconds, highlighting the crucial role of the module in handling long-duration speech.

**Q5: Does Entity Dictionary Really Alleviate the Problem of Difficult Entity Extraction?** To further explore how the proposed Entity Dictionary helps facilitate the task, we carry out an analysis study on the ACE05-EN⁺ dataset. We first count the frequency distribution of all the occurrences of entities in the train set, as shown

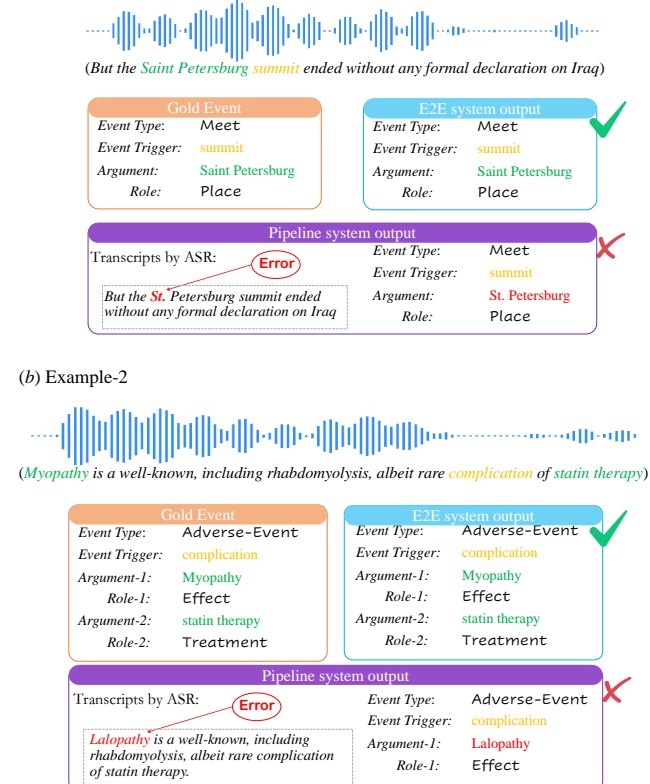

**Figure 9: Qualitative examples of pipeline method and E2E method.**

in Fig. 7, and we find that the distribution of entities is characterized by an obvious long-tailed distribution. Such data distribution characteristic makes the model tend to ignore the knowledge of uncommon entities during training and the model has difficulty generating words it rarely sees. However, the Entity Dictionary helps focus on low-frequency entity information in the data source by introducing external knowledge. The component of the Entity Dictionary is mainly names of people, organizations, and places as shown in Fig. 8, which alleviates mistakes such as homophones, incorrectly extracting people's name "*Emmalie*" as "*Emily*" where the former is rarely seen while the latter is more common.

## 6.5 Qualitative Case Study

Finally, we provide a more intuitive understanding of the differences in performance between our pipeline and E2E systems on specific instances by offering some qualitative case studies. We randomly select two samples from the sentence-level test set, where our E2E model correctly produced outputs that matched the gold events for both instances. However, the pipeline model fails in both cases, demonstrating typical errors. For example-1, the pipeline system incorrectly recognized "*Saint Petersburg*" as "*St. Petersburg*" during the ASR stage (due to biases in the training of the ASR model). This error propagates through the system, leading to incorrect identification of the argument in the subsequent EE step. For example-2, similarly, the ASR mistakenly identifies the word "*yopathy*" as "*Lalopathy*", which results in incorrect event argument

outcomes. Additionally, constrained by the two-step prediction paradigm, the pipeline system only identifies one argument, failing to recognize the second argument.

## 7 What To Do Next with SpeechEE?

We believe firmly that the proposed SpeechEE will open a new era for the multimodal IE community. Here we shed light on the potential directions for future research.

• **Mitigating Noise Impact.** Speech in real scenarios always includes background noise and other types of interference. Our experimental results also indicate that noisy backgrounds impose additional challenges on SpeechEE. We believe it is promising to develop stronger mechanisms to help filter out ambient noise in speech, enhancing task performance.

• **Identifying Implicit Elements.** Beyond noise issues, SpeechEE often encounters implicit elements. While most EE results can find corresponding audio segments in speech, sometimes words are swallowed or not explicitly pronounced (termed as implicit elements). Compared to explicit elements, identifying implicit ones poses a greater challenge. We consider it crucial to devise smarter methods to address the recognition of these implicit elements.

• **Cross-language SpeechEE.** Our dataset includes two major languages, English and Chinese, with annotations that are not parallel across languages. Future research can explore cross-lingual transfer learning in speech, investigating the role of language-invariant features in enhancing EE task.

• **Weak/Unsupervised SpeechEE.** In this paper, we primarily focus on supervised learning using a substantial amount of annotated data. We deem it essential to leverage our benchmark for weak or unsupervised SpeechEE. Some current multimodal large language models (MLLMs) [15, 16, 55, 56, 62, 64] already exhibit significant unsupervised generalization capabilities. Future research can explore weak or unsupervised approaches in SpeechEE.

• **Better Evaluation Metric for SpeechEE.** The current evaluation method for the EE task strictly matches predicted event records with the golden label. However, given that the input for the SpeechEE task is purely audio without textual information, strict matching significantly hinders performance. A new evaluation metric that accommodates fuzzy semantic matching is expected to be proposed for a fair evaluation of SpeechEE. For example, an entity that semantically matches the core meaning with the gold label should be considered correct.

## 8 Conclusion

In this paper, we introduce a novel task, SpeechEE, extracting structured event information from speech. We first contribute a comprehensive dataset tailored to this task, which features diverse scenarios, languages, domains, and speaker styles, constructed from both synthesis and human reading. Further, we propose an E2E SpeechEE model to offer a strong baseline for the task. Through analysis, we demonstrate the complexity of the task, and the effectiveness of our approach. Finally, as pioneers in this topic, we highlight key directions for future research.

## Acknowledgments

This work is supported by the National Natural Science Foundation of China (NSFC) Grant (No. 62336008).

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
