# OpenReview forum: "SpeechEE: A Novel Benchmark for Speech Event Extraction"
_acmmm.org/ACMMM/2024/Conference — MM2024 Poster_

### Official Review · Reviewer_qYRh · 2024-04-27

**Rating:** 5
**Confidence:** 3

**Summary:**

The paper introduces SpeechEE, a novel benchmark for extracting structured event information from speech. It addresses a significant gap in the field of information extraction by creating a large-scale, high-quality dataset and proposing an end-to-end (E2E) model to handle the complexities inherent in speech event extraction. The authors claim that their work provides a strong baseline for future research and outlines key directions for advancement in the field.

**Strengths:**

The paper tackles a novel and important problem in information extraction from speech, which has been under-explored.
The authors have constructed a comprehensive dataset with a diverse range of scenarios, languages, and domains, which is a significant contribution to the field.
The proposed E2E model includes innovative components such as a Shrinking Unit module and a retrieval-aided decoding mechanism, which are well-motivated and technically sound.
The experimental results are extensive and demonstrate the efficacy of the proposed model across various subsets of the SpeechEE dataset.

**Limitations:**

The paper could benefit from a more detailed discussion on the limitations of the current approach and potential areas for improvement.
While the dataset is comprehensive, the authors might consider the inclusion of more languages and dialects to enhance the model's generalizability.

**Suitability:**

3

---

### Official Review · Reviewer_65GM · 2024-05-22

**Rating:** 3
**Confidence:** 3

**Summary:**

This paper a new task named SpeechEE which aims to achieve event extraction from speech waveforms. A dataset is constructed and an end-to-end model is proposed for this task.

**Strengths:**

1. The task is new.
2. A dataset is constructed with variaties on language, domain and scenario.

**Limitations:**

1. The constructed datasetset has its limitations. It developed from text EE datasets and the scripts for human narration and TTS are in the format of written language, which can't cover the characteristics of spoken language in the practial scenarios of SpeechEE>
2. A retrival-aided text decoder is designed with an entity dictionary. How to deal with the entities that are unseen in the training dataset?
3. Although the experimental results show the E2E method outperformed the pipeline method, the limitations of the E2E method is not fully discussed. One significant limitation is the dependency on supervised training data. While, for the pipeline method, it can use plenty of speech recogntion and text EE datasets to improve the performance of each module.

**Suitability:**

3

---

### Official Review · Reviewer_yuiZ · 2024-05-26

**Rating:** 4
**Confidence:** 3

**Summary:**

This paper pioneers SpeechEE, defined as detecting the event predicates and arguments from a given audio speech, by constructing a large-scale high-quality dataset based on textual EE datasets, manually by real-person narration and automatic synthesis. Also, authors build E2E SpeechEE system based on encoder-decoder architecture to show that the proposed shrinking unit module and retrieval-aided decoding works in diverse scenarios.

**Strengths:**

- This paper constructs the dataset for SpeechEE task by manual narration and automatic audio synthesis, featuring diverse scenarios and domains.
- Shrinking unit module and retrieval-aided decoding are devised for the strong baseline of the task.

**Limitations:**

- Though diverse language usages was featured, English and Chinese were used and not as a parallel corpus, so that the detailed comparison is only available in ACE05 datasets.
- It is still not clear why SpeechEE is more desired compared to TextEE, given that the performance of E2E SpeechEE is marginally higher than the pipeline and also that the pipeline score would depend on the performance of ASR module. (+ It would have been nice if the speechEE covered multi-party speaking scenarios).

**Suitability:**

2

---

### Meta-Review · Area_Chair_tugK · 2024-06-30

**Recommendation:** Accept (Poster)
**Confidence:** 3

**Metareview:**

The submitted paper presents a novel approach to information extraction from speech. The paper introduces innovative methodologies and constructs a comprehensive dataset that significantly contributes to the field. Below is a detailed summary of the strengths and limitations highlighted by the reviewers.

Strengths:
The paper constructs a robust dataset for the SpeechEE task through manual narration and automatic audio synthesis, incorporating diverse scenarios and domains.
The paper introduces a Shrinking Unit module and a retrieval-aided decoding mechanism, which are well-motivated and technically sound. These components strengthen the proposed end-to-end (E2E) model.
The SpeechEE task itself is novel and addresses an important, under-explored problem in the field of information extraction from speech.
The experimental results are extensive and demonstrate the efficacy of the proposed model across various subsets of the SpeechEE dataset. The paper shows a clear performance improvement of the E2E model over baseline methods.

Limitations:
Although the dataset features diverse language usages, it currently includes only English and Chinese, and these are not provided as a parallel corpus. Reviewers noted that it remains unclear why SpeechEE is preferred over TextEE, given the marginal performance improvement of the E2E SpeechEE over the pipeline method, which depends on the performance of the ASR module. Additionally, the dataset does not cover multi-party speaking scenarios.
The dataset is developed from text EE datasets, and the scripts for human narration and TTS are in written language format, which may not capture the full characteristics of spoken language in practical SpeechEE scenarios.
While the E2E method outperformed the pipeline method, the paper does not fully discuss the limitations of the E2E approach, particularly its dependency on supervised training data. In contrast, the pipeline method can leverage extensive speech recognition and text EE datasets to improve performance.

The paper presents a significant contribution to the field of information extraction from speech through its novel task, comprehensive dataset, and innovative methodologies. While there are some limitations, they do not undermine the overall quality and potential impact of the work. The identified limitations provide valuable directions for future research rather than critical flaws in the current submission.

Given the constructive nature of the feedback provided by the reviewers, I recommend that this paper be accepted for presentation at the conference. The authors are encouraged to address the reviewers’ feedback in their final revision to further enhance the paper's contributions and impact.